# Inflammation and Immunity Gene Expression Patterns and Machine Learning Approaches in Association with Response to Immune-Checkpoint Inhibitors-Based Treatments in Clear-Cell Renal Carcinoma

**DOI:** 10.3390/cancers15235637

**Published:** 2023-11-29

**Authors:** Nikolas Dovrolis, Hector Katifelis, Stamatiki Grammatikaki, Roubini Zakopoulou, Aristotelis Bamias, Michalis V. Karamouzis, Kyriakos Souliotis, Maria Gazouli

**Affiliations:** 1Department of Basic Medical Sciences, Laboratory of Biology, Medical School, National and Kapodistrian University of Athens, Michalakopoulou 176, 11527 Athens, Greece; ndovroli@med.uoa.gr (N.D.); katifel@med.uoa.gr (H.K.); matgr@med.uoa.gr (S.G.); 22nd Propaedeutic Department of Internal Medicine, ATTIKON University Hospital, School of Medicine, National and Kapodistrian University of Athens, 11527 Athens, Greece; rzakopoul@gmail.com (R.Z.); abamias@med.uoa.gr (A.B.); 3Molecular Oncology Unit, Department of Biological Chemistry, National and Kapodistrian University of Athens, 11527 Athens, Greece; mkaramouz@med.uoa.gr; 4School of Social and Education Policy, University of Peloponnese, 22100 Corinth, Greece; info@ksouliotis.gr; 5Health Policy Institute, 15123 Athens, Greece

**Keywords:** ccRCC, immunotherapy, TKIs, machine learning, cancer

## Abstract

**Simple Summary:**

Clear cell renal cell carcinoma (ccRCC) is the most common form of renal cancer. Currently, treatment of metastatic ccRCC (mccRCC) is challenging despite the use of modern immunotherapy options. In order for the patient to receive the most effective treatment among the available pharmaceutical agents, there is a constant need for biomarkers that can predict therapeutic efficacy. The present study investigates changes in the mRNA expression of genes related to inflammation and immunity in patient blood. By using machine learning approaches, several changes in mRNA expression levels were observed in patients who had clinical benefit from the treatment compared to patients with progressive disease. Collectively, our results show that gene expression can be used to classify these samples with high accuracy and specificity.

**Abstract:**

Clear cell renal cell carcinoma (ccRCC) is the most common renal cancer. Despite the rapid evolution of targeted therapies, immunotherapy with checkpoint inhibition (ICI) as well as combination therapies, the cure of metastatic ccRCC (mccRCC) is infrequent, while the optimal use of the various novel agents has not been fully clarified. With the different treatment options, there is an essential need to identify biomarkers to predict therapeutic efficacy and thus optimize therapeutic approaches. This study seeks to explore the diversity in mRNA expression profiles of inflammation and immunity-related circulating genes for the development of biomarkers that could predict the effectiveness of immunotherapy-based treatments using ICIs for individuals with mccRCC. Gene mRNA expression was tested by the RT2 profiler PCR Array on a human cancer inflammation and immunity crosstalk kit and analyzed for differential gene expression along with a machine learning approach for sample classification. A number of mRNAs were found to be differentially expressed in mccRCC with a clinical benefit from treatment compared to those who progressed. Our results indicate that gene expression can classify these samples with high accuracy and specificity.

## 1. Introduction

Renal cell carcinoma (RCC) represents the most common renal malignancy, accounting for more than 80% of all renal cancer cases, with clear cell RCC (ccRCC) being the most common subtype (>80% of all cases) [1]. When detected at an early stage, disease is curable with surgery in most cases [2]. Nevertheless, almost 1 out of 4 patients presents with metastatic disease, while 30–40% of patients presenting with localized cancer will develop metastases after nephrectomy [3].

The prognosis of metastatic ccRCC (mccRCC) has been significantly improved during the last 15 years due to the introduction of agents that target essential mechanisms of ccRCC development, expansion, and aggressiveness. These mainly include angiogenesis via inhibition of the tyrosine kinase (TKIs) of the Vascular Endothelial Growth Factor Receptor (VEGFR) and an anti-tumor immune response by blocking interactions, which result in the suppression of immune response against the tumor [4]. Contemporary immunotherapy of mccRCC includes two distinct types of molecules: programmed cell death-1 (PD-1) inhibitors and cytotoxic T-lymphocyte-associated protein 4 (CTLA-4) inhibitors. PD-1 is a glycoprotein found in a variety of cells, including T cells. When its ligand (PD-L1), which can be overexpressed by tumor cells, binds to PD-1, it inhibits the proliferation of PD-1+ cells and promotes tumor evasion. PD-1 inhibitors (e.g., Nivolumab) and mediate enhanced antitumor activity [5]. CTLA-4 is a receptor found in T cells that exhibits inhibitory effects. Its downstream targets include cell cycle machinery molecules, which are essential for cell cycle progression [6].

The introduction of modern immunotherapy in our armamentarium greatly improved the prognosis of mccRCC [7,8]. However, not all patients benefit from a given regimen. In addition, the optimal sequence of novel agents for each patient remains to be found. Currently, a challenge for personalized oncology remains the identification of biomarkers capable of predicting response/resistance to treatment, which at the same time will be applicable in the clinical practice. One of the first biomarkers to be investigated in patients who received ICI treatment was PD-L1. Its expression in the tumor microenvironment has been associated with responses to immunotherapy in a variety of tumors [9,10]. Nevertheless, this is not the case for mccRCC [11]. In addition, obtaining sufficient tumor samples from all patients with advanced disease can be practically challenging. Thus, serum markers represent a more realistic option for everyday practice [12].

Recent studies have shown that the expression of immune-related genes has provided a link between tissue expression and a patient’s outcome. These include increased tumor expression of *CCR7* that indicates poor prognosis in mccRCC patients treated with TKIs [13] and increased *CXCR4* [7], *CXCL5* [8], *CCL4* [9], *CCL22* [10], and *IL-1* [11], which have been linked to poor outcomes by promoting high proliferation rates/aggressive phenotypes. On the contrary, high *IDO-1* expression is associated with favorable response status to immunotherapy [12]. Whether the altered gene expressions observed in the tumor microenvironment are reflected in the serum or peripheral blood samples remains an important yet unanswered question. Currently, there are considerably fewer studies that have identified mRNA expression changes in immune-related genes in patient blood samples. In terms of transcriptomic alterations in blood, down-regulated *IL-18R1* and *IL-18RAP* are associated with treatment response in advanced lung cancer [14], while a model using pretreatment mRNA levels of 15 genes in melanoma has been proposed for the prediction of immunotherapy outcome [15]. Regarding mccRCC, studies that utilize serum mRNA with regards to immunotherapy response aside from *PD-1* are scarce and include upregulation of *IL-9R*, which signifies response to treatment, and upregulation of *RETN*, which indicates lack of response in immunotherapy-treated mccRCC patients [16,17]. Moreover, at the protein level, increased serum IL-1α, IL-6, CCL44, and IL-13 are correlated with increased survivorship in mccRCC patients that receive immunotherapy [18].

In the present study, we investigated the predictive value of distinct molecular signatures of human cancer inflammation and immunity genes that could serve as circulating biomarkers for the prediction of the response of mccRCC patients to ICIs-based therapeutic approaches. In addition, we aim to elucidate the cascade of transcriptomic and immunological changes these treatments attain beyond their intended targets, along with any changes enacted by different therapeutic options.

## 2. Materials and Methods

### 2.1. Patients

Peripheral blood samples from mccRCC patients (*n* = 27) were acquired before treatment administration. All participants received anti-PD1 agents (Nivolumab/Pemprolizumab) in combination with anti-CTLA-4 (Ipilimumab) or TKIs (Cabozantinib/Axitinib) as first-line therapy for metastatic disease. The best response to treatment was categorized into: clinical benefit (CB) [complete response (CR) or partial response (PR) or stable disease (SD)] or progressive disease (PD) using the RECIST criteria (v.1.1) [19]. Participants were treated at Attikon General University Hospital and Laikon General Hospital between 2021 and 2023. All subjects involved in this study provided their written informed consent prior to participating, and the present study was approved by the Ethics Committee of the hospitals. The clinicopathological data are summarized in Table 1, while all individual data are described in detail in Appendix A.

### 2.2. Differential Gene Expression Analysis (DGEA)

Total peripheral blood RNA was extracted using the Qiagen AllPrep RNA/DNA Mini Kit (Qiagen, Hilden, Germany). cDNA was prepared using the RT^2^ First Strand Kit (Qiagen) according to the manufacturer’s instructions. Gene expression quantification was performed by RT^2^ profiler PCR Array human cancer inflammation and immunity crosstalk (PAHS-181ZA, Qiagen) using the RT^2^ qPCR SYBR Green Master Mix (Qiagen).

Comparative analysis involved dividing our samples according to CB and PD, or timing in relation to treatment initation, and performing the following comparisons: (A) patients with CB (control) and PD mccRCC at baseline to detect biomarkers able to predict their response to immunotherapy; (B) patients with CB before (control) and after therapy; (C) patients with PD before (control) and after immunotherapy; (D) patients with PD and CB (control) disease after immunotherapy; and (E) patients regardless of disease progression that receive either combinations of ICI/ICI or ICI/TKI (control) treatments. The last three comparisons were conducted to allow for investigating the transcriptional differences exacted by the therapy. Data analysis was conducted using the RT^2^ Profiler PCR Array Data Analysis version 3.5 from Qiagen. All samples successfully passed quality checks for PCR Array reproducibility, RT efficiency, and genomic DNA contamination.

For within-sample normalization, five housekeeping genes (*ACTB*, *B2M*, *GAPDH*, *HPRT1*, and *RPLP0*) were utilized with the 2^−ΔCt^ method. The fold change (FC) was calculated as 2^−ΔΔCt^ and is presented as fold regulation in the results. Genes that were underexpressed are represented as the negative inverse of the fold change, while overexpressed genes are presented as the fold change. The *p*-values were determined by conducting Student’s *t*-tests on the replicate 2^−ΔCt^ values for each gene in both the control and test groups. Only genes with fold regulation higher than 2 or less than −2 and a *p*-value of less than 0.05 are reported for the A, B, and C groups, while groups D and E disregard *p*-value restrictions due to the small sample size and the scope of the analysis. A graphical representation was created using R on the 1/ΔCt values to both normalize and showcase higher expression levels. The analysis was performed on the Qiagen Geneglobe platform (found online at https://geneglobe.qiagen.com/, accessed on 20 October 2023). Ct values for each gene per sample were provided in separate excel files, which are provided as Appendix A)). 

### 2.3. Machine Learning (ML)

To enhance and complement differential expression findings as predictive biomarkers for disease progression and effects of therapy, H_2_O AutoML (v.3.40.0.4) [20], a machine learning automation framework, was used. We performed predictive modeling tasks on the previously defined (A–C) subsets of samples, in which each sample subset was divided into two parts with a 60/40 split, one used for training and one for validation. Both sets contained samples of both classes under investigation (CB and PD at baseline, CB before and after therapy, and PD before and after therapy). Unfortunately, subsets D (PD or CB after therapy) and E (ICI/ICI versus ICI/TKI after therapy) did not contain enough samples for a sufficient ML analysis. H_2_O AutoML was then applied to the training dataset using a 10-fold cross-validation approach for each fold iteratively, training and tuning a total of 50 different models on various algorithms and hyperparameter combinations, excluding ensemble and deep learning algorithms. The top 5 best-performing models in total were selected based on their respective performance metrics, area under the curve (AUC), and root mean squared error (RMSE). After completing the training process, each of the 5 models was tested against the separate training set to evaluate their performance and generalizations on new data. In addition, the top 10 contributing features for each model were gathered and reported, along with the algorithm and the performance metrics for each model. This has allowed for both feature selection and generalization of the models for the classification of new samples. The R script that was used is provided in the Appendix A.

## 3. Results

Both the DGEA and ML approaches highlighted several genes that define specific transcriptomic signatures with the potential to predict which patients are more likely to respond well to treatment, as well as provide insights into how their immunological profiles change after the intervention. The ML algorithms highlighted here, as expected, even though favored genes with higher expression differences also provided more in-depth insights into our subgroupings, highlighting transcriptomic patterns without the restrictions of multiple testing and taking into account more complex relationships between genes and metrics such as distribution.

### 3.1. Prediction of Response to Therapy

When comparing patients with known outcomes at baseline, we were able to identify upregulated and downregulated genes that can act as potential biomarkers of how effective treatment might be for each group. In total, 19 genes are differentially expressed in the studied groups. Patients with PD exhibited high upregulation of *ACKR3* (Atypical Chemokine Receptor 3, also known as *CXCR7*) with a 36.1-fold regulation, making it a prime candidate for further investigation, as it is considered an “orphan” chemokine with no known ligand. In addition, *BCL2* (B-Cell Lymphoma 2), a critical anti-apoptotic protein that inhibits cell death with a 22.8-fold change, appears to contribute to mccRCC cell survival. Other genes, such as *SPP1* (Secreted Phosphoprotein 1), *CCL22* (C-C Motif Chemokine Ligand 22), *GBP1* (Guanylate-Binding Protein 1), *FASLG* (Fas Ligand), *MICA* (MHC Class I Polypeptide-Related Sequence A), *PTGS2* (Prostaglandin-Endoperoxide Synthase 2), and *CTLA4* (Cytotoxic T-Lymphocyte-Associated Protein 4), also displayed moderate upregulation with fold changes ranging from 5.0 to 7.7. Additionally, slight increased mRNA expression (4.9–2.7-fold regulation) in cases with progressive disease was observed for the genes *CCL28* (Chemokine (C-C motif) ligand 28), *CCL4* (C-C Motif Chemokine Ligand 4), *IRF1* (Interferon Regulatory Factor 1), and *IL12A* (Interleukin 12A). On the other hand, *CXCR1* (C-X-C Chemokine Receptor Type 1), which is involved in regulating neutrophil migration and activation, showed significant downregulation with a 12.5-fold change in its expression. In addition, *CD274* (Programmed Cell Death 1 Ligand 1, also known as *PD-L1*), *FOXP3* (Forkhead Box P3), *CCL21* (C-C Motif Chemokine Ligand 21), *CSF1* (Colony-Stimulating Factor 1), and *CXCL11* (C-X-C Motif Chemokine Ligand 11) exhibited a profile more closely associated with patients that will respond to therapy. Figure 1A depicts these transcriptomic differences.

When the same data were processed through our ML pipeline, the five top models managed to achieve AUCs of 0.8–1 with low RMSE values. Gradient boosting machine (GBM) algorithms and their variation, the XGBOOST (extreme gradient boosting), with various hyperparameters, appear to perform well for this kind of analysis occupying the top four spots, albeit with variable accuracies. The fifth model highlighted was a GLM (generalized linear model) algorithm, which required more features than the rest of the models in order to successfully classify our samples. At least four out of five models agree on *CSF1*, *CXCR1*, *STAT1* (Signal Transducer and Activator of Transcription 1), *CCL18* (C-C motif Chemokine Ligand 18), and *ACKR3.* In addition, *CSF1* appears to be the strongest biomarker candidate, performing very well on all models. All the models and the top 10 genes used for distinguishing between patients who will respond to treatment are featured in Figure 1B. 

### 3.2. Transcriptomic Changes in CB after Therapy

To better understand how the transcriptional profile of ccRCC patients who will respond to immunotherapy changes after treatment we employed the same approaches as before. In total, 15 genes appear to be differentially expressed after the intervention. *TNF* (Tumor Necrosis Factor), whose role is triggering immune reactions and maintaining tissue homeostasis, exhibited an impressive 94.8-fold-regulation increase after treatment. *IRF1* (Interferon Regulatory Factor 1) also displayed a considerable upregulation of 35.5-fold suggesting an upregulation in the production of interferons. Other significantly upregulated genes were *HIF1A* (Hypoxia-Inducible Factor 1 Alpha) with a 28.1-fold regulation, the cytotoxic *GZMB* (Granzyme B) by 24.0-fold, *FOXP3* by 16.3-fold, *TLR3* (Toll-Like Receptor 3) by 15.4-fold, *CCL28* (C-C Motif Chemokine Ligand 28) by 14.8-fold. *CXCL11* by 13.1-fold, the tissue healing *IGF1* (Insulin-like Growth Factor 1) by 11.7-fold and *IL4* (Interleukin 4) by 10.7-fold. The effects of treatment were also showcased in the downregulation of *PCDP1* (Programmed Cell Death Protein 1, PD-1) by 20.9-fold and *CXCL10* (C-X-C Motif Chemokine Ligand 10), by 5.0-fold. All the relevant results are shown in Figure 2A.

As previously, GBM ML models performed very well in highlighting key molecules before and after treatment in patients with CB. All five top models exhibited high accuracy with AUCs of 1 and low RMSE scores. In addition to the aforementioned GBM models the DRF (Decision Rule-based Fuzzy) and the GLM algorithms utilized more genes to classify the groupings effectively. In the comprehensive analysis, it was revealed that at least 4 out of the 5 models placed significant emphasis on certain key genes. These molecules were identified as *CCL18*, *GZMB*, *CCL28*, *HLA-A* (Human Leukocyte Antigen-A), and prominently *CSF3* (Colony-Stimulating Factor 3). All molecules that emerged as possible biomarkers for effectiveness of treatment and the respective algorithms are depicted in Figure 2B. 

### 3.3. Transcriptomic Changes in PD after Therapy

Identifying patients whose treatment might prove ineffective and having a short timeline for transitioning to another therapeutic strategy is very important. For this reason, we studied the effects of intervention on patients who did not respond to their current regiments before and after treatment. From the DGEA analysis, we observed significant upregulation of several genes associated with immune activation and inflammatory responses after treatment. *GZMB* exhibited a 39.7-fold increase, while *FOXP3*, *TNFSF10* (Tumor Necrosis Factor Superfamily 10), and *IFNG* (Interferon Gamma) showed fold upregulations of 39.0, 19.8, and 19.7, respectively. Furthermore, *IL4*, *HLA-A*, *VEGFA* (Vascular Endothelial Growth Factor A), *TLR3*, *CCXCL11*, *MIF* (Macrophage Migration Inhibitory Factor), and *CXCR1* all demonstrated moderate upregulation, with fold changes ranging from 13.7 to 4.8. In contrast, we observed a notable downregulation of *TGFB1* (Transforming Growth Factor Beta 1) after treatment, with a fold regulation of −10.7. Figure 3A contains all the pertinent results for the total of 12 differentially expressed genes detected.

Once more, ML algorithms were used to further detect genes that distinguish the transcriptomic profiles of patients with progressive mccRCC before and after treatment. A mixture of GBM, DRF, and XRT (extremely randomized trees) algorithms performed as the top 5 models in the ML approach. All achieved an AUC of 1, with low RMSEs utilizing a variety of features. At least 4 out of 5 algorithms agree on *GZMB*, *CSF3*, *IL4*, *IFNG*, and highlight the role of *FOXP3*. Since *CSF3* and *GZMB* were also highlighted in the analysis of the CB patients, we can safely assume that their contribution is an effect of the treatment itself and does not contribute to the early detection of treatment response. Figure 3B illustrates the molecules identified as potential biomarkers for treatment effectiveness, along with the corresponding algorithms used in the analysis.

### 3.4. Transcriptomic Changes Affected by Therapy in PD and CB Patients

To supplement our analyses and further elucidate differences between responders and non-responders to therapy, we analyzed the expression differences in patients with a progressive course of disease versus those that appear to be more stable at follow-up. We chose to examine differences in all genes that had statistically significant dysregulation after therapy when compared to the baseline samples (analyses on subgroups B and C). The analysis revealed significant changes in the regulation of several genes. Those that exhibited substantial upregulation include *PDCD1* (74.37-fold), *HLA-A* (60.76-fold), suggesting enhanced antigen presentation to T cells and *TNFSF10* (44.22-fold). *VEGFA* (18.48-fold), *CXCL10* (13.15-fold), and MIF (9.03-fold). Meanwhile, *CXCR1*, *TGFB1*, and *IFNG* also show a moderate upregulation pattern with a fold regulation of 3.67-fold, 1.53-fold, and 1.41-fold, respectively. 

On the other hand, genes that show modest downregulation include *CCL28* (−1.64-fold), *FOXP3* (−1.67-fold), *GZMB* (−1.75-fold), *TLR3* (−1.97-fold), and *IL4* (−2.13-fold). *IGF1*, *CCL21*, *CXCL5*, *CSF1, CXCL11*, *IRF1*, and *TNF* show a higher downregulation with −4.95, −6.92, −11.25, −12.13, −12.70, −14.67, and −14.76 folds, respectively. The most significant downregulation was observed in *HIF1A* (−43.71-fold). All results are presented in Figure 4.

### 3.5. Transcriptomic Changes Affected by Different Therapeutic Combinations

In addition to previous analyses, DGEA was conducted to examine the influence of the ICI/ICI and ICI/TKI combinatorial treatments on the expression levels of inflammation and immunity genes. For the analysis, we used the ICI/TKI-treated patients as a baseline and reported on the top 10 upregulated and downregulated genes found in the ICI/ICI group. The results revealed significant fold regulation changes, although we failed to achieve statistical significance due to the low number of samples and the variation between individuals. Notable upregulated genes include *TP53*—Tumor Protein p53 (9.3-fold), *IL1A*—Interleukin 1 Alpha (3.7-fold), *ACKR3* (3.3-fold), *HLA-A* (3.3-fold), *BCL2L1*—BCL2 Like 1 (2.8-fold), *CCR9*—C-C Motif Chemokine Receptor 9 (2.8-fold), *KITLG*—KIT Ligand (2.8-fold), *NOS2*—Nitric Oxide Synthase 2 (2.8-fold), *MIF* (2.5-fold), and *CXCL8*—Interleukin-8 (2.5-fold). On the other side, downregulated genes include *CXCL5* (−2.4-fold), *TNF* (−2.4-fold), *CSF1* (−2.5-fold), *EGFR*—Epidermal Growth Factor *Receptor* (−2.5-fold), *TGFB1* (−2.5-fold), *CD274* (−2.8-fold), *CSF2*—Colony Stimulating Factor *2* (−2.9-fold), *HIF1A* (−3.0-fold), *IL15*—Interleukin 15 (−4.0-fold), and *EGF*—Epidermal Growth Factor (−5.1-fold). 

All results are presented in Figure 5. The complete DGEA results, which conform to the constraints of statistical significance we imposed for all subgroupings, can be found in Table 2.

## 4. Discussion

ICIs-based therapies have revolutionized cancer therapeutic strategies; however, the responses of mccRCC patients and survival rates are still poor [20]. Additionally, reliable biomarkers for therapy response prediction and individualized patient selection for proper therapy are still limited. In the present study, we investigated how the expression of inflammation and immune gene signatures in the peripheral blood prior to ICI-based therapy differs in patients with CB and PD and how those are affected by therapy. 

The pretreatment gene signature we identified contains 19 differentially expressed genes from the initial 84 tested for classifying patients according to response. *ACKR3* mRNA was significantly overexpressed in progressive disease. *ACKR3* is overexpressed in many human cancers, including renal carcinoma, and is correlated with poor prognosis, tumor progression, and metastasis [21,22]. It is important to notice that atypical chemokine receptors (ACKRs) are basic regulatory components of the chemokine network in a wide range of pathological conditions, including cancer, suggesting the need to validate them as novel targets for immunotherapies [23]. Furthermore, we identified elevated levels of *BCL2* mRNA expression in cases of progressive disease. Although there are discrepancies in the literature regarding its expression in renal cancer [24,25], it is established that heightened expression of this gene in tumors correlates with unfavorable responses to systemic cancer therapy and a resistance to immunotherapy [26,27]. Other genes that had increased expression in progressive disease cases were *SPP1*, *CCL22*, *GBP1*, *FASLG*, *MICA*, *PTGS2*, *CTLA4*, *CCL28*, *CCL4*, *IRF1*, and *IL12A.* Our findings are in agreement with previous studies. In particular, *SPP1* expression demonstrated increased levels in the progressing group and decreased levels in the regressing group within an RCC cohort, underscoring its potential as an indicator in immunotherapy [28], while high expression of *PTGS2* mRNA is a poor prognostic indicator in human mccRCC [29]. Additionally, high *FASLG* expression in RCC is associated with a significantly worse prognosis [30], and FasL neutralization has the potential to improve the efficacy of immunotherapy based on immune checkpoint inhibitors. The effect of *GBP1* expression in cancers appears to be complex. Zhao et al. [31] based on the TCGA database report increased expression of *GBP1* in kidney renal clear cell carcinoma cases, and by examining 33 cancer types, they suggested that generally, patients with high *GBP1* expression may possibly result in better responses to immunotherapy; however, other studies like the one by Ye et al. [32] show that *GBP2,* which is highly co-expressed with *GBP1*, is a robust prognostic biomarker for high immune infiltration and poor prognosis in ccRCC. In our cohort, we observed that GBP1 has a moderate increase in cases with progressive disease. It is well recognized that many human cancers, including ccRCC, express the MHC class I chain-related polypeptide A (MICA) protein that serves as a ligand for the activating NK group 2D (NKG2D) receptor on NK cells [33]. Torres et al. [34] reported that targeting MICA-expressing tumors with an anti-MICA antibody significantly delayed their growth, and more recently, Secchiari et al. [35] reported that higher expression of *MICA* was associated with worsened overall survival for ccRCC patients. Our results align with these observations since we found increased *MICA* expression in progressive disease patients. Regarding *CTLA4*, it is well known that it is the second target of checkpoint inhibition therapeutic approaches in RCC, and *CTLA4* expression was significantly correlated with metastatic diseases and associated with a reduced survival in ccRCC [36]. We found moderate upregulation of *CTL4* expression in cases with progressive disease at baseline; however, recently, the findings of Klumper et al. [37] indicated that reduced methylation of the *CTLA4* promoter, linked to elevated mRNA levels, predicts a positive response to immune checkpoint blockade and favorable outcomes among ccRCC patients. This positive effect offsets the initial negative prognostic impact of *CTLA4* hypomethylation. However, it is worth noting that *CTLA4* methylation appears to be specifically predictive for immunotherapy and is not correlated with the response to TKIs. This observation helps shed light on our results, given that our cohort includes patients undergoing a combination of immunotherapy and TKI treatments. It has also been shown, consistent with our findings, that high levels of chemokines *CCL4*, *CCL22*, and *CCL28* in pre-treatment tumor specimens were associated with worse patient overall survival after immunotherapy in RCC [38,39,40]. We also observed a small increase in *IRF1* and *IL12A* mRNA in the progressive disease group. Recently, Chehrazi-Raffle et al. [41] have also reported that mRCC patients with lower levels of circulating IL12 have a clinical benefit during treatment with ICI or TKI. Regarding IRF1 in mccRCC, the data from Chen et al. [42], primarily sourced from public databases, suggests that cases with a low IRFscore (constructed based on the transcriptomic expression of the IRF family) may exhibit heightened sensitivity to targeted therapies, while those within the high IRFscore subgroup might be more responsive to immunotherapy. In addition, they highlight the correlation of a low IRFscore with the mccRCC2/3 phenotypes, while a high IRFscore matched with mccRCC1/4. Thus, our results can be partially justified since our cohort includes patients that received combined therapies but are also agnostic to the subphenotyping of mccRCC, highlighting the need for further study to determine biomarker viability. 

Corro et al. [43] suggested that *CXCR1* expression is associated with cancer stem cell-like properties of ccRCC as well as negatively correlated with overall survival. As far as we know, there are no studies examining the *CXCR1* expression levels and the immunotherapy response, but our results highlight an upregulation of *CXCR1* mRNA in patients with PD versus those with CB after therapy, while before therapy the phenomenon is reversed. These results appear to be in agreement with a study by Panaiyadiyan et al. [44], which suggested that increased tumor *CXCR1* expression before TKIs relates to progression-free survival and can predict reduced benefit of therapy in patients with mccRCC. However, it is known that *CXCR1* is associated with a direct migration of neutrophils and neutrophil infiltration in several cancers and is implicated in both anti- and pro-tumor roles [45]. Thus, more studies are needed to elucidate the role of *CXCR1* in modulating the immunotherapy response. 

Concerning the *CD274* expression in mRCC, its possible predictive value for response to immunotherapy is still controversial, and the results from the analyses of the clinical trials investigating ICIs in this disease are inconclusive [46]. However, in agreement with Kang et al. [47], we observed that elevated *CD274* mRNA expression at the baseline was significantly associated with CB patients. *FOXP3* mRNA was also upregulated in baseline in the patient group that will exhibit stable disease, but also after therapy. Interestingly, Koh et al. [48] suggested that higher levels of FoxP3+ Treg cells can predict a beneficial response to anti-PD-1 immunotherapy in patients with advanced non-small cell lung cancer, and this observation can rationalize our results. Patients with CB in our cohort were found to have increased *CCL21* and *CXCL11* mRNA levels. In preclinical mouse models, *CCL21* was shown to facilitate antitumor activity via recruiting and colocalizing NK cells, DCs, and T cells in tumors, and it was supported that anti-PD-1 administered in combination with CCL21-DC tumor antigen therapeutic vaccines eradicated lung cancer [49]. Regarding *CXCL11*, its mRNA upregulation was associated with a better prognosis in several cancers, and PD-L1 blockade combined with an oncolytic vaccinia virus expressing *CXCL11* in mouse models was shown to considerably decrease tumor burden and improve prognosis [50,51]. In our results, we also found moderately elevated mRNA *CSF1* baseline levels in the CB patient group as well as after therapy. Our findings seem to be in contrast with studies supporting the idea that high levels of *CSF1* and tumor-associated macrophages (TAMs) are associated with a poor cancer prognosis [52] and that inhibition of *CSF-1/CSF-1R* signaling can improve the efficacy of checkpoint blockade in animal tumor models by enhancing anti-CTLA-4 and anti-PD-1-induced tumor immunity [53]. Nevertheless, it remains controversial whether *CSF-1/CSF-1R* signaling basically functions through regulating tumor immunity or tumor cell malignancy. Studies indicated that *CSF-1R* is mainly expressed in tumor cells, while CSF-1R has also been reportedly expressed in tumor-associated macrophages (TAMs) and involved in tumor immune escape [54,55]. Becht et al. [56] identified 4 robust ccRCC subtypes (ccRCC1 to ccRCC4). The immunome identified the ccRCC4 subgroup as exhibiting the highest expression of *CSF1* and proposed that the ccRCC4 subgroup identifies patients that may respond to therapeutic immune checkpoint modulators. These observations indicate that therapeutic responses to immunotherapy are multifaceted and can be influenced by a wide range of factors, including the specific tumor type, the overall immune landscape, and the genetic and molecular characteristics of the tumor. In general, the case of *CSF1* is a curious one. Proposed by our ML approaches as a high-confidence feature for sample classification, it appears to have a pleiotropic role during ccRCC. On one hand, it has a clear homeostatic role in cell proliferation and survival and has been found elevated in healthy subjects [57], and on the other, during cancer, it is implicated in M2 macrophage polarization, which promotes cancer cell proliferation and cell infiltration [58]. Its immunological role both enhances and hinders cancer progression, either by assisting said TAM formation or by actively contributing to the fight against cancerous cells. Even the role of TAMs is not definite since there appears to be a similar dichotomy in their role, both promoting proliferation and killing cancer cells [59,60]. It appears that the role of *CSF1* and TAMs is heavily reliant on ccRCC subphenotypes and a very delicate balance in expression, which when perturbed, can lead in opposing directions [59]. Ongoing research aims to unravel these complexities and identify reliable biomarkers and predictors of immunotherapy response are needed. 

ICI-based treatment has a profound effect on gene expression in our cohort. In patients that responded to therapy, we found 15 differentially expressed genes before and after therapy. Among them, *TNF* was found to be significantly overexpressed after treatment. It is known that ICI treatment is associated with increased *TNF* gene expression, and in agreement with our findings, it has been reported that melanoma patients responding to ICI had higher gene expression of *TNF* and TNF response signatures after therapy compared to non-responders [61]. Additionally, we also found increased TNF expression in CB patients when we compared the post-therapy gene expression between CB and PD cases. The pattern of upregulation extends to other immune-related genes as well, albeit at a lower rate, suggesting a strong immune response overall and seemingly training the immune system to fight off mccRCC progression. As expected, *PDCD1,* the gene that encodes PD-1, was significantly downregulated after successful anti-PD-1 therapy. PD-1/PD-L1 blockade has shown prolonged survival benefits in several malignancies [62]. Regarding *CXCL10*, increased transcription for *CXCL10* mRNA has been observed after immunotherapy treatment (i.e., nivolumab) [63]. However, overexpression of CXCL10 could enhance RCC cell metastasis [64]. Thus, its downregulation following treatment seems to be consistent with CB, as shown by our results when comparing PD patients versus CB after therapy. 

Regarding the patients with progressive disease, *GZMB* was found to be upregulated post-treatment compared to pre-treatment samples. Au et al. [65] have shown a small increase in *GZMB* expression post-nivolumab treatment in both responders and non-responders, but CD8+T cell-specific *GZMB* expression was to be found significantly increased in responders. When we compared post-therapeutic gene expression, we also found increased *GZMB* expression in patients with CB compared to patients with PD. Nevertheless, it is supported that in spite of the favorable outcome associated with *GZMB* expression in tumors, its expression in some cases was associated with poor prognosis, resistance to therapy, and advanced cancer stage [66]. Additionally, we observed increased levels of *FOXP3*, *TNFSF10*, and *IFNG* following treatment compared to pre-treatment samples in progressive disease group patients. Jensen et al. [67] reported that IL-2-based therapy leads to the accumulation of FOXP3-positive immune cells in the tumor microenvironment in metastatic renal cell carcinoma and that high numbers of on-treatment FOXP3-positive cells were correlated with poor prognosis. Tumor necrosis factor-related apoptosis-inducing ligand (*TRAIL*) is encoded by *TNFSF10*, and high *TRAIL* expression levels are linked with poor disease-specific survival in patients with RCCs [68]. Furthermore, even if the mechanism is unclear, interferon-γ (IFNG)-mediated adaptive resistance is one basic barrier to improving immunotherapy in solid tumors [69]. Although *TGFB1* exhibited downregulation after treatment in the progressive disease group, its expression remained elevated compared to the CB group. It is well known that TGF-β is a multifunctional cytokine that acts in a cell- and context-dependent way as a tumor promoter or suppressor, and its pleiotropic nature contributes to drug resistance, tumor escape, and a weakened response to therapy [70]. Several TGF-β-targeting therapies are under clinical investigation in combination with anti-PD-(L)1 therapies, particularly in tumor types that have had poor responses to anti-PD-(L)1 monotherapies. 

The expression analysis of post-therapy samples further highlighted differences between responders and non-responders. Most of these changes have been discussed previously, but some results were accentuated. For example, *HLA-A* overexpression in progressive disease not only was highlighted as distinctive between PD and CB after therapy but appears to also be promoted by the interventions under study in the PD group, hinting at a synergistic interaction. HLA-A has been proposed as a therapeutic target for ccRCC in recent studies [71,72], in addition to studying specific defects that promote immune escape to tumors contrary to their typical role [73]. High expression of VEGFA in patients with progressive disease after therapy is also quite common since it is a marker and facilitator of highly vascularized tumors [74] and the target of the TKI-based therapies used in our study. HIF1A was significantly upregulated in CB patients after therapy when compared both to their baseline and to progressive disease patients. In line with our results, HIF1A has been proposed as an effective biomarker for response to immunotherapy [75] and as a cancer suppressor gene in ccRCC [76]. 

Finally, the evaluation of ICI/ICI versus ICI/TKI using differential expression analysis revealed some interesting findings. First and foremost, the ICI/ICI combination appears to upregulate the expression of the onco-suppressor *TP53* gene that encodes the p53 protein. Even though rarely-occurring mutations of *TP53* have been linked to poor prognosis in ccRCC [77] and its overexpression in ccRCC has not shown any real clinical potential [78], its elevated expression is under-reported and demands further study. The fact that the combined ICI/ICI therapy might induce expression of this gene over the ICI/TKI approach can be deemed inconclusive, or it can also be a marker of cell overcompensation and immune system overdrive. Such is also the case of *IL1A*, which appears to have a dual role in cancer [79], and in ccRCC especially, its blocking has been associated with better outcomes [80]. On the other hand, *EGF* and *EGFR* overexpression found in the ICI/TKI group has once more been implicated in silico with poorer outcomes [81] and presents another controversial answer to our questions since EGFR in ccRCC has been shown to be on the uptake even after EGFR-targeted therapies due to a variety of tumor-specific characteristics [82]. At this junction, it is important to note that our analysis contained both responders and non-responders to therapy, which might be influencing these results, and the low number of samples in each group prevented results from reaching statistical significance regardless of their seemingly perturbed expressions. The fact that ICI and TKI treatments have different targets further complicates the extraction of any conclusions for this analysis, even though in literature ICI/TKI combinations are presented as more effective [83] or at best requiring further investigation [84]. 

To enhance the DGEA analysis results, we employed ML approaches in both conventional and unconventional ways. In the case of predicting response to therapy at baseline, ML, in its traditional role, classified our samples, highlighting *CSF1*, *CXCR1*, *STAT1*, *CCL18*, and *ACKR3* as important features that promote said classification. While this knowledge does not fully reveal the optimal expression levels for each feature, it does provide a distinct pointer toward smaller feature subsets that should be investigated collectively. This in tandem study can enable the accurate classification of new samples based on their potential to respond to therapy or not. Hence, we deemed it appropriate to also apply the same logic in a nonconforming way to let ML models select important features that discern transcriptional changes effected by therapeutic intervention. This approach can serve a dual purpose: aiding upcoming research in pinpointing features that precisely signal therapeutic response on a molecular scale and providing clinicians with insights into potential risks associated with maintaining therapy, even when a patient’s clinical status remains uncertain at a specific timepoint. These results might be able to serve as the basis for future works, especially those conducted with larger sample sizes, which would be able to provide more robust results. 

## 5. Conclusions

Our research, like many others in the field, faces several significant limitations due to the inherent characteristics of the specific cancer under investigation and the immunotherapy strategies employed. These limitations are crucial to acknowledge and consider when interpreting our findings. First and foremost, one of the major challenges we encounter is the low incidence rate of the cancer of interest, ccRCC, within the general population. This rarity makes it difficult to assemble a sufficiently large and diverse cohort of participants for our study, especially since this is a single-center study. Globally, the prevalence of ccRCC is estimated to be less than 2% [85]. As a result, our study may suffer from a limited sample size, which may not accurately represent the broader population affected by this cancer. The scarcity of available samples is further compounded by the fact that ccRCC primarily affects individuals later in life, and those who do not respond well to treatment often have a very poor prognosis. Consequently, we have access to only a small number of post-treatment samples for comparison, which in turn diminishes our statistical power to detect meaningful trends or draw definitive conclusions. Additionally, the complexity of our research is heightened by the diversity of treatment approaches undertaken by the participating patients. Our patients were treated with combinations of ICIs and TKIs, each targeting different molecular pathways. This diversity in treatment strategies introduces a layer of complexity to our study, as these therapies can have distinct effects on the immune system and tumor microenvironment, although this makes the study relevant to everyday clinical practice. The interplay between various molecules as treatment targets and the resulting immunological cascade effects further complicates our ability to elucidate clear and concise outcomes, prompting further validation of these results. In summary, our research confronts significant limitations stemming from the rarity of ccRCC, the resulting challenges in sample accrual, the limited post-treatment samples available, and the intricate nature of the immunotherapy strategies employed. While we strive to overcome these limitations to the best of our abilities, it is essential to approach our findings with a nuanced understanding of these constraints and their potential impact on the generalizability and robustness of our conclusions.

Addressing the limitations associated with the rarity of clear cell renal cell carcinoma (ccRCC), challenges in sample accrual, limited post-treatment samples, and complex immunotherapy strategies requires a multi-faceted approach. Collaborative research efforts can be instrumental, as they allow for the pooling of resources, samples, and data from multiple research institutions or centers that specialize in ccRCC. Additionally, collaborating with patient registries and advocacy groups focused on ccRCC can help identify potential participants and access existing datasets, thereby providing a larger and more diverse study cohort. To overcome the challenge of small sample sizes, longitudinal studies can also be considered. These studies involve following patients over an extended period, which can lead to the collection of more post-treatment samples over time, ultimately enhancing the statistical power of the research. In these prospective studies, patients receiving specific immunotherapy regimens are enrolled and monitored from the outset, which can reduce the complexity introduced by different treatment histories and increase the homogeneity of the study population. Additionally, computational modeling and simulation studies can complement empirical research by exploring the potential effects of different treatment strategies and outcomes under varying conditions. This can also allow us to investigate the complex immunological interactions associated with different immunotherapy approaches, where the use of additional in vitro or animal models may be considered. Where the sample size allows it, patient stratification based on specific criteria, such as treatment history or genetic markers, can help mitigate the variability introduced by different treatment approaches. 

Regardless of these constraints, this study validates and expands current knowledge on the impact of immunotherapy in ccRCC and whether we can safely predict patient response, even though not all immunological mechanisms can be effortlessly explained. The baseline predictive biomarkers highlighted within were detected by easily accessible and non-invasive blood draws, giving clinicians a possibly fast and consistent way to assess patient response to selected regiments. By relying on routine blood tests, healthcare practitioners can swiftly ascertain whether the immunotherapy is yielding the desired effect, enabling timely adjustments to optimize patient outcomes and paving the way for enhanced patient care and personalized treatment strategies. The importance of identifying the right marker to support clinical decision-making in RCC goes beyond the potential (predictive) impact on treatment outcomes and carries economic repercussions as well. This is why, across cancer care, studies underline the importance of careful evaluation of both their clinical utility and their cost-effectiveness before wide-spread adoption in routine clinical practice [86]. The discussion is not new—the call to assess the impact on cost of care of introducing predictive biomarkers has been intensifying over the past decade [87]. Still, there is very poor literature on the matter. This is mainly due to the limited research horizon, which focuses on the clinical and cost impact of a treatment after its selection and fails to combine the probabilities to choose that treatment upfront, which may be affected using a biomarker. This results in a fragmentation of the evaluation of the cost-effectiveness impact of a biomarker on the total patient pathway and disallows a much-needed holistic approach to inform decision-making, including reimbursement.

Our research, which has identified potential predictive biomarkers of response to treatment through the integration of differential expression analysis and machine learning techniques, has the potential to significantly impact the field of personalized medicine and patient care. By tailoring therapies based on an individual’s genetic or molecular profile, healthcare providers can increase treatment efficacy while minimizing adverse effects. Patients are more likely to receive treatments that are specifically targeted to their unique biological characteristics, increasing the chances of a positive response and better disease management. Biomarker-driven treatment decisions can reduce the administration of ineffective treatments to non-responsive patients. This not only spares patients from unnecessary side effects but also optimizes healthcare resources and reduces healthcare costs. While promising, the integration of biomarkers and machine learning also presents challenges, including the need for rigorous validation, potential bias in data sources, and ethical considerations regarding patient privacy and data sharing.

## Figures and Tables

**Figure 1 cancers-15-05637-f001:**
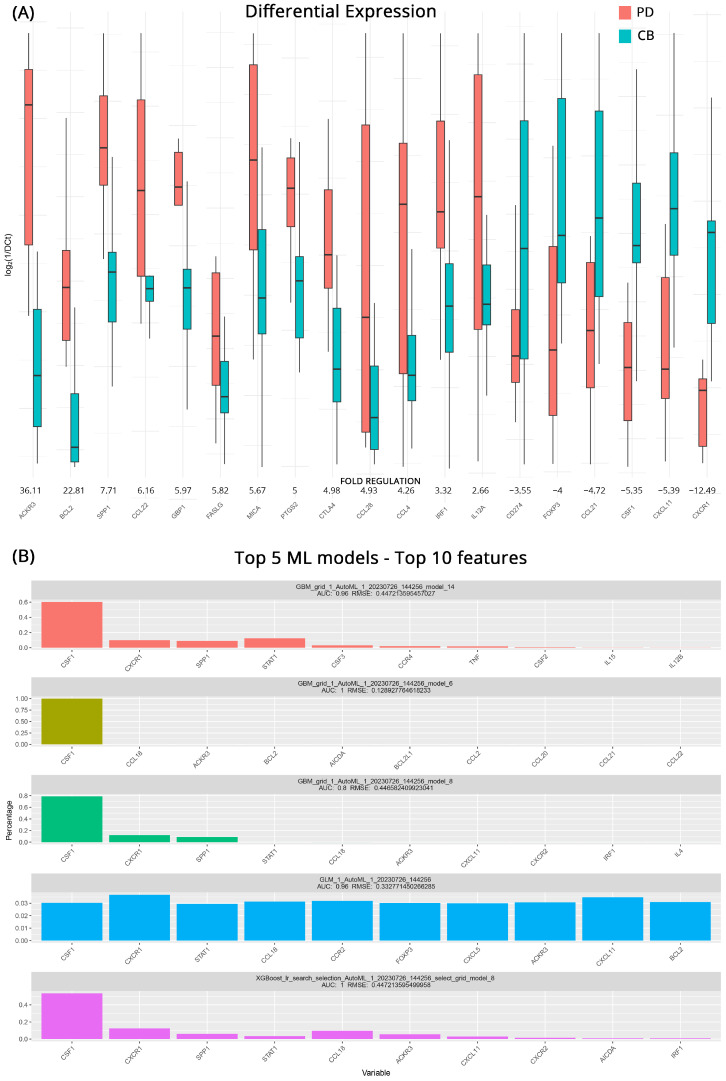
(**A**) Differential gene expression results at baseline (pre-therapy) for patients with clinical benefits (CB) and progressive disease (PD) (subgroup A). Bar charts showcase differences in expression based on log_2_ΔCt values for each group. Fold regulations for each gene highlight their differences after applying the 2^−ΔΔCt^ method. Negative fold regulation signifies downregulation in the progressive disease group. (**B**) Top 5 ML models that best classify the samples to either group. For each one of them, the most important features as well as their name, AUC (Area Under the Curve), and RMSE (Root Mean Square Error) values are displayed.

**Figure 2 cancers-15-05637-f002:**
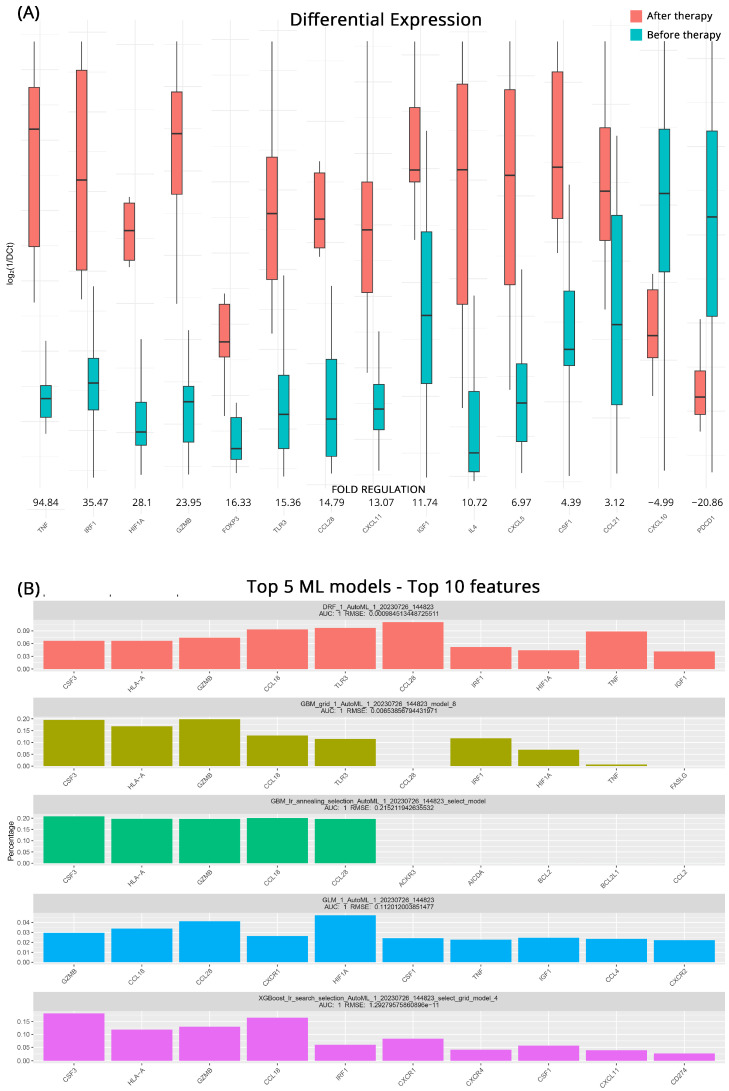
(**A**) Differential gene expression results for patients with Clinical Benefits (CB) before and after therapy (subgroup B). Bar charts showcase differences in expression based on log_2_ΔCt values for each group. Fold regulations for each gene highlight their differences after applying the 2^−ΔΔCt^ method. Negative fold regulation signifies downregulation in the patients after therapy. (**B**) Top 5 ML models that best classify the samples into either group. For each one of them the most important features as well as their name, AUC (Area Under the Curve), and RMSE (Root Mean Square Error) values are displayed.

**Figure 3 cancers-15-05637-f003:**
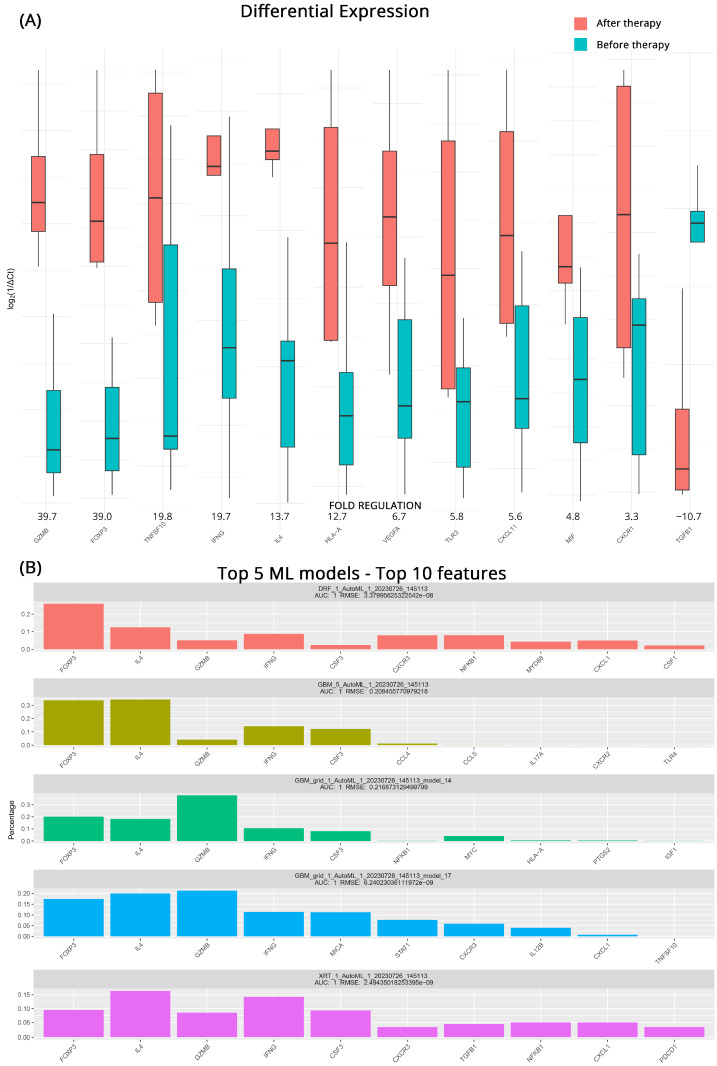
(**A**) Differential gene expression results for patients with progressive disease (PD) before and after therapy (subgroup C). Bar charts showcase differences in expression based on log_2_ΔCt values for each group. Fold regulations for each gene highlight their differences after applying the 2^−ΔΔCt^ method. Negative fold regulation signifies downregulation in the patients after therapy. (**B**) Top 5 ML models that best classify the samples into either group. For each one of them, the most important features as well as their name, AUC (area under the curve), and RMSE (root mean square error) values are displayed.

**Figure 4 cancers-15-05637-f004:**
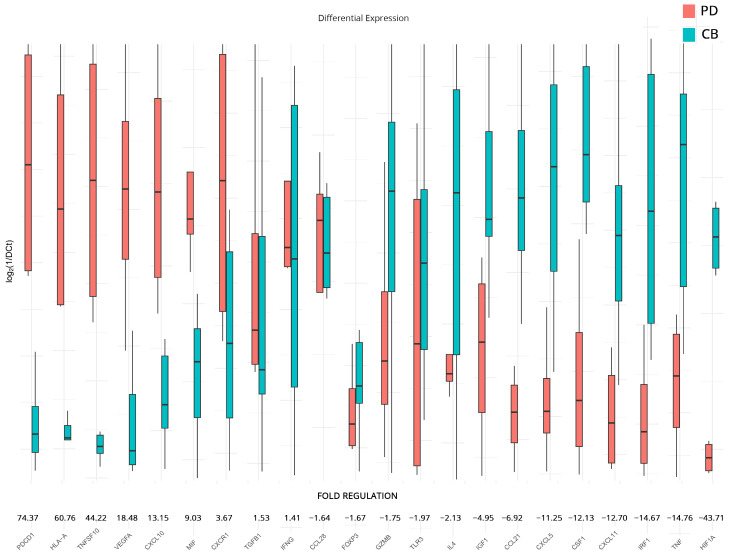
Differential gene expression results after therapy for patients with clinical benefits (CB) and progressive disease (PD) (subgroup D). Bar charts showcase differences in expression based on log_2_ΔCt values for each group. Fold regulations for each gene highlight their differences after applying the 2^−ΔΔCt^ method. Negative fold regulation signifies downregulation in the progressive ccRCC group.

**Figure 5 cancers-15-05637-f005:**
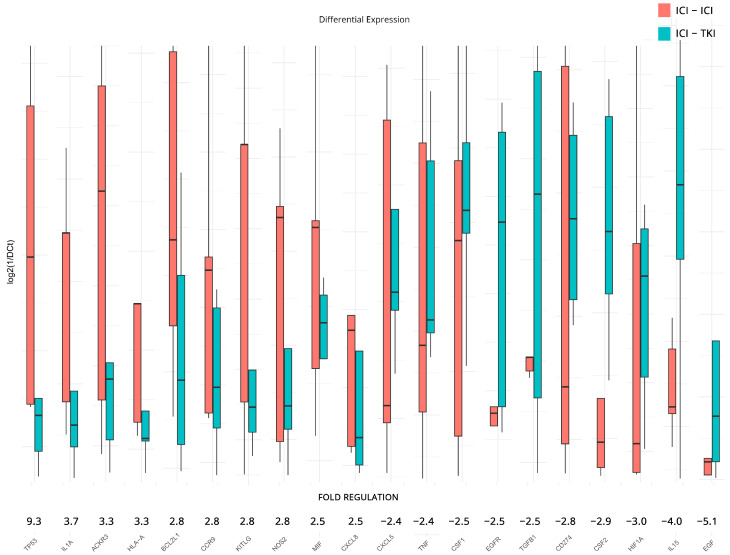
Differential gene expression results after therapy for patients under combination of ICI/ICI or ICI/TKI treatments. Bar charts showcase differences in expression based on log_2_ΔCt values for each group. Fold regulations for each gene highlight their differences after applying the 2^−ΔΔCt^ method. Negative fold regulation signifies downregulation in the ICI/ICI group.

**Table 1 cancers-15-05637-t001:** Patient clinicopathological data.

Characteristic		Number of Patients (%)
**Gender**	Male	21 (77.78%)
	Female	6 (22.22%)
**Median age (Range)**		66.4 ± 10.7
**Treatment**	Nivolumab + Ipilimumab	11 (40.74%)
	Nivolumab + Cabozantinib	8 (29.63%)
	Pemprolizumab + Axitinib	8 (29.63%)
**Response Status**	Clinical Benefit (CB)	14 (54.85%)
	Progressive Disease (PD)	13 (48.15%)

**Table 2 cancers-15-05637-t002:** DGEA results for all subgroups.

PD vs. CB at Baseline			PD vs. CB after Therapy			ICI/ICI vs. ICI/TKI after Therapy		
Gene Symbol	Fold Regulation	*p*-Value	Gene Symbol	Fold Regulation	*p*-Value	Gene Symbol	Fold Regulation	*p*-Value
*ACKR3*	36.1	>0.001	*PDCD1*	74.4	0.030	*TP53*	9.3	0.310
*BCL2*	22.8	0.026	*HLA-A*	60.8	0.050	*IL1A*	3.7	0.884
*SPP1*	7.7	>0.001	*TNFSF10*	44.2	0.036	*ACKR3*	3.3	0.377
*CCL22*	6.2	0.016	*VEGFA*	18.5	0.041	*HLA-A*	3.3	0.783
*GBP1*	6.0	0.013	*CXCL10*	13.2	0.045	*BCL2L1*	2.8	0.183
*FASLG*	5.8	0.031	*MIF*	9.0	0.055	*CCR9*	2.8	0.287
*MICA*	5.7	0.019	*CXCR1*	3.7	0.053	*KITLG*	2.8	0.553
*PTGS2*	5.0	0.034	*TGFB1*	1.5	0.678	*NOS2*	2.8	0.901
*CTLA4*	5.0	0.050	*IFNG*	1.4	0.747	*MIF*	2.5	0.242
*CCL28*	4.9	0.023	*CCL28*	−1.6	0.676	*CXCL8*	2.5	0.966
*CCL4*	4.3	0.017	*FOXP3*	−1.7	0.376	*CXCL5*	−2.4	0.942
*IRF1*	3.3	0.023	*GZMB*	−1.8	0.227	*TNF*	−2.4	0.905
*IL12A*	2.7	0.017	*TLR3*	−2.0	0.659	*CSF1*	−2.5	0.747
*CD274*	−3.6	0.026	*IL4*	−2.1	0.121	*EGFR*	−2.5	0.751
*FOXP3*	−4.0	0.025	*IGF1*	−5.0	0.097	*TGFB1*	−2.5	0.110
*CCL21*	−4.7	0.005	*CCL21*	−6.9	0.034	*CD274*	−2.8	0.952
*CSF1*	−5.4	0.004	*CXCL5*	−11.3	0.046	*CSF2*	−2.9	0.487
*CXCL11*	−5.4	0.005	*CSF1*	−12.1	0.026	*HIF1A*	−3.0	0.782
*CXCR1*	−12.5	0.008	*CXCL11*	−12.7	0.049	*IL15*	−4.0	0.052
			*IRF1*	−14.7	0.043	*EGF*	−5.1	0.244
			*TNF*	−14.8	0.050			
			*HIF1A*	−43.7	0.030			
**CB after vs. before Therapy**			**PD after vs. before Therapy**					
**Gene Symbol**	**Fold Regulation**	***p*-Value**	**Gene Symbol**	**Fold Regulation**	***p*-Value**			
*TNF*	94.8	>0.001	*GZMB*	39.7	>0.001			
*IRF1*	35.5	0.001	*FOXP3*	39.0	>0.001			
*HIF1A*	28.1	>0.001	*TNFSF10*	19.8	0.045			
*GZMB*	24.0	>0.001	*IFNG*	19.7	0.004			
*FOXP3*	16.3	0.003	*IL4*	13.7	>0.001			
*TLR3*	15.4	0.003	*HLA-A*	12.7	0.008			
*CCL28*	14.8	0.003	*VEGFA*	6.7	0.031			
*CXCL11*	13.1	0.001	*TLR3*	5.8	0.008			
*IGF1*	11.7	0.003	*CXCL11*	5.6	0.008			
*IL4*	10.7	>0.001	*MIF*	4.8	0.014			
*CXCL5*	7.0	0.001	*CXCR1*	3.3	0.010			
*CSF1*	4.4	0.001	*TGFB1*	−10.7	0.028			
*CCL21*	3.1	0.031						
*CXCL10*	−5.0	0.039						
*PDCD1*	−20.9	0.039						

## Data Availability

The data generated and/or analyzed during the current study are available from the corresponding author on reasonable request.

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
