# Peer review of "Inflammation and Immunity Gene Expression Patterns and Machine Learning Approaches in Association with Response to Immune-Checkpoint Inhibitors-Based Treatments in Clear-Cell Renal Carcinoma"

_cancers, 2023, doi:10.3390/cancers15235637_

Round 1

Reviewer 1 Report

Comments and Suggestions for Authors

The manuscript by Dovrolis et.al. is a resubmission of their previous work which required major improvements, however the major change in the paper is not clearly described. The improvement should be in terms of addressing the shortcoming of their previous submission. They should provide a point-by-point discussion about the insufficiency of their previous submission and the endower they took to address those comments. Suggestion for completing the manuscript:

The major problem of this manuscript arises from unsubstantiated assumption that blood samples from different parts of the body can represent tumor microenvironment and can be a good indicator of immune check point response.

My suggestion is (1) to verify this assumption from other cohorts, if available, or increase the number of samples and discuss a rigorous power analysis to indicate that these genes are the only genes that can be found and they are not a subset of other genes that are only found by chance and (2) to develop a biological model that can verify their assumption.

Comments on the Quality of English Language

No comments.

Author Response

Our answers are attached. Thank you for your comments

Reviewer 2 Report

Comments and Suggestions for Authors

After my suggestion was to be rejected, the authors tried to convince me. Thanks for that. More information provided  but the problem still persists; Totally, 27 patients were enrolled in this study. The sample size was insufficient, which weakened the persuasion of the results.

Author Response

(The authors gave the same response as above.)

Reviewer 3 Report

Comments and Suggestions for Authors

Thanks for answering the questions. I accept your manuscript.

Author Response

Thank you for your kind comments

Round 2

Reviewer 1 Report

Comments and Suggestions for Authors

I understand the difficulty of the need for updating the protocol and I hope in your next step you analyze the tumor microenvironment blood samples.

Comments on the Quality of English Language

NA